

# Developing a quality indicator system for evaluating internet plus home care nursing services based on the SERVQUAL model: a Delphi-analytic hierarchy process study

Lei Ye[1], Shulan Yang[1], Biyan Jiang[1], Caixia Liu[1], Xiaoqing Jin[2] and Polun Chang[3]

[1] Department of Nursing, Zhejiang Hospital, Hangzhou, China
[2] Department of Acupuncture, Zhejiang Hospital, Hangzhou, China
[3] Institute of Biomedical Informatics, Yang-Ming Chiao-Tung University, Taipei, Taiwan

Corresponding author
Shulan Yang, 947373396@qq.com

## ABSTRACT

**Background:** In the context of global population aging and the rapid development of information technology, the demand for Internet Plus Home Care Nursing (Internet + HCN) services have been on the rise, especially in China. Internet+ HCN services have the potential to maximize existing human resources to counter the shortage of medical healthcare services. However, at present, Internet+ HCN services are difficult to scale due to the lack of standardized service quality governance. Quality indicators for service evaluation of Internet+ HCN services are under-defined.

**Objective:** To develop a quality indicator system for evaluating Internet+ HCN services, and to shed theoretical light on assessing mHealth service quality from a user experience perspective.

**Methods:** An initial quality indicator system was established based on scenarios related to Internet+ HCN services. The Delphi Method was applied to modify the indicators according to experts' opinions, and the analytic hierarchy process (AHP) was applied to calculate the indicator weight. Finally, a quality indicator system for evaluating Internet+ HCN services was developed based on the SERVQUAL model.

**Results:** Altogether, 17 experts from relevant fields such as nursing education, clinical nursing, health management, and health informatics were consulted through email surveys. The response rates in both rounds of Delphi and the AHP were 100%. The average expert authority coefficients were 0.912 and 0.925 respectively in the two rounds of Delphi. Kendall's W, indicating variation coefficients, ranged from 0.262 to 0.265. Finally, a quality indicator system for evaluating Internet+ HCN services, comprising five primary indicators and 15 secondary indicators, was developed. Primary indicators and their AHP generated the following weights: assurance (0.245), reliability (0.240), tangibles (0.192), responsiveness (0.190), and empathy (0.132).

**Conclusions:** By measuring the services quality gap between user expectations and perceptions, the proposed SERVQUAL model-based quality indicator system shows potential in improving the quality of Internet+ HCN services through the perspective of user experiences.

# INTRODUCTION

The demographic shift towards an aging global population has exacerbated the shortage of medical healthcare resources. According to data released by the *World Bank (2022)* and *National Bureau of Statistics of the People's Republic of China (2021)*, the proportion of the population aged 65 and above in the European Union, America, and China were 21%, 17%, and 13.50% respectively in 2020. It indicated the transition into a hyper-aged, aged, and aging society in these regions. However, the number of registered nurses per 1,000 people was strikingly low, with America at 9.8, the European Union at 7.9, and China at only 3.34.

The elongation of life expectancy, coupled with the high prevalence of chronic diseases, poses a formidable challenge to aging societies. Numerous studies, including those by *Mitchell & Walker (2020)*, *Partridge, Deelen & Slagboom (2018)*, *Maresova et al. (2019)*, and *Bialystok et al. (2016)*, have highlighted how global aging exacerbates the burden of chronic diseases. This, in turn, intensifies the demand for home care nursing services (*Healthy China Action Promotion Committee, 2019*) and further strains healthcare resources, as documented by *Keevil & Romero-Ortuno (2015)*, *Marć et al. (2019)*, *Abdi et al. (2019)*, and *Aaltonen & van Aerschot (2021)*. In the new era of "Internet+," the implementation of internet-based services may have the potential to alleviate this pressure by maximizing existing human resources.

In previous studies exploring the application of internet-based interventions in healthcare, specifically within the nursing domain, studies have examined the positive impact on health outcomes across various populations (*Akinosun et al., 2021*; *Sherifali et al., 2018*; *Lin et al., 2020*). The interventions have demonstrated potential to improve lifestyle behaviors, physical health, and psychological well-being. In China, an innovative internet-based home care nursing service has been booming in recent years (*National Health Commission of the People's Republic of China, 2019*), which is called Internet Plus Home Care Nursing (Internet+ HCN). This service refers to home care nursing services provided by qualified home care nurses from medical institutions supported by integrated online services. Theoretically, Internet+ HCN could both bridge the gap in care needs (*Zhao et al., 2021*; *Qiu & Qiu, 2022*) and optimize healthcare resource deployment (*Zimansky et al., 2018*). To narrow the gap between service demand and supply, China spares no effort in advocating Internet+ HCN services (*Beerens et al., 2014*; *Morris et al., 2013*) to maximize and optimize home care nursing supply in virtue of the booming informatics technology.

By integrating online and offline services, nurses from medical institutions are empowered to provide viable care for their discharged patients and other high-home care nursing demand clients in the community. Nevertheless, in practice, the development of Internet+ HCN services is just passable in line with expectations in China (*Zhao et al., 2021*). Previous researches have identified some implementation barriers in internet-based nursing services (*He et al., 2024*; *Tian et al., 2023*). For nurses, these barriers include insufficient training, time constraints, and concerns regarding liability and personal safety.

On the patient side, challenges such as limited internet access, low computer literacy, and trust in online interventions have been identified as key obstacles. Generally for Internet+ HCN services, standing problems include insufficient nurses involved in the service pool, unknown risks underlying the process of providing home services, and the lack of standardized service quality governance. Quality indicators for service evaluation of Internet+ HCN services are under-defined. Moreover, services quality directly influences the practice and promotion of Internet+ HCN services, as emphasized by *Wang et al. (2021b, 2022a, 2022b)*. Thus, it is of great significance to develop a quality indicator system for evaluating Internet+ HCN services and to form a long-term evaluation mechanism to promote sustainable development of the services.

The present study selected the SERVQUAL model as the theoretical basis for constructing a quality evaluation indicator system. The SERVQUAL model is a multidimensional research instrument designed to measure service quality by detecting gaps between users' expectations and perceptions (*Fan et al., 2017*; *Feng et al., 2022*; *Parasuraman, Zeithaml & Berry, 1985*). Using this model, this study compared the gap between users' expectation and perception to help realize both the objective evaluation and prediction of their satisfaction towards Internet+ HCN services. Portions of this text were previously published as part of a preprint (DOI 10.2196/preprints.48711).

# METHODS

## Initial potential indicators

User experience is critical to the quality of Internet+ HCN services. The underlying digitization, integration, and modernization features of the services could cater to users' diversified, differentiated, and quality-oriented services demand. A simplified version of the SERVQUAL model (*Parasuraman, Zeithaml & Berry, 1985*), which is composed of Reliability, Assurance, Tangibles, Empathy, and Responsiveness, also known as the RATER model, was adopted in developing the initial potential indicators. The RATER model has been widely used in service quality management (*Czaplewski, Olson & Stanley, 2002*). The five dimensions of the RATER model (reliability, assurance, tangibles, empathy, and responsiveness) align well with the characteristics of Internet+ home care nursing services, making it effective in assessing the gap between user expectations and perceptions of service quality. Based on the model, reliability represents the ability of performing the service effectively and accurately. Assurance refers to the skill of producing trust and the credibility of the users, which requires proper knowledge and service awareness. Tangibles represent tangible services such as physical facilities, equipment, personnel, and communication material. Empathy refers to the attention and priority that the service gives to meet the needs and demands of its users. Responsiveness is related to the ability and willingness to assist the user and provide the appropriate service as promised. All five dimensions were defined as primary indicators, while secondary indicators were developed according to scenarios related to current practices in Internet+ HCN services.

A literature search for evidence discussing relevant potential quality indicators of evaluating Internet+ HCN services was undertaken, using comprehensive database of biomedical literature both in English and Chinese, such as PubMed, CINAHL and CNKI

*etc.*, from 2012 to 2022. The primary search term was "Internet Plus Nursing", and related terms or synonyms which would be used in various combinations were also searched, including "Telehealth Nursing", "e-Nursing", "Mobile Health Nursing", "Digital Nursing" *etc*. The initial literature search yielded a total of 3,213 articles. After removing those not relevant to quality management, 82 articles were thoroughly reviewed to extract potential indicators specific to the quality management of Internet+ HCN services. The results of this search contributed to a comprehensive understanding of the current state of research in this field. Based on the review, the initial indicators were developed through group discussions within the research team, and were assorted into the five dimensions of the RATER model. Meanwhile, semi-structured interviews were performed to bring the experiences and perceptions of service providers and users together to optimize the indicators. A total of 10 individuals were interviewed, including five service providers and five users of Internet+ HCN services. These interviews aimed to gather valuable insights into the experiences and perceptions of both groups and key themes and findings were identified to inform the refinement of the indicator system. Finally, an initial indicator system consisting of five primary indicators (reliability, assurance, tangibles, empathy, and responsiveness) and 15 secondary indicators was developed.

It is worth mentioning that users' experience as well as their expectations towards the service affect the perceived service quality. Users were therefore requested to rate both their perception and expectation towards the service when the proposed evaluation system was being implemented. The difference between user expectation and perception defines the service quality gap. Figure 1 shows the theoretical framework of the proposed quality indicator system for evaluating Internet+ HCN services based on the SERVQUAL model.

## Expert selection

The sample of experts for the present study was recruited through a strategic process designed to ensure a representative panel. Recruitment efforts were multifaceted, targeting individuals with specialized knowledge and experience relevant to the objective of developing a quality indicator system for evaluating Internet+ HCN services in the context of China. Researchers who had a proven track record of research, publications, or professional practice in the targeted field are identified potential experts, drawing from academic institutions, professional organizations, and industry groups. To broaden the reach and ensure a diverse range of perspectives, the initial group of experts nominated by the researchers was then asked to suggest other qualified individuals who could contribute valuable insights to the study. The selection of experts was guided by a predefined set of criteria as follows: (1) having more than 5 years work or research experience relevant to Internet+ HCN services in their professional area; (2) being familiar with Internet+ HCN services; (3) volunteering to participate in this study and in several rounds of consultation communication. Prospective panel members were provided with detailed information about the study, its objectives, the expected time commitment, and the confidentiality measures in place. They were required to provide informed consent before being included in the panel. And a written consent was achieved by electronic form at the beginning of the

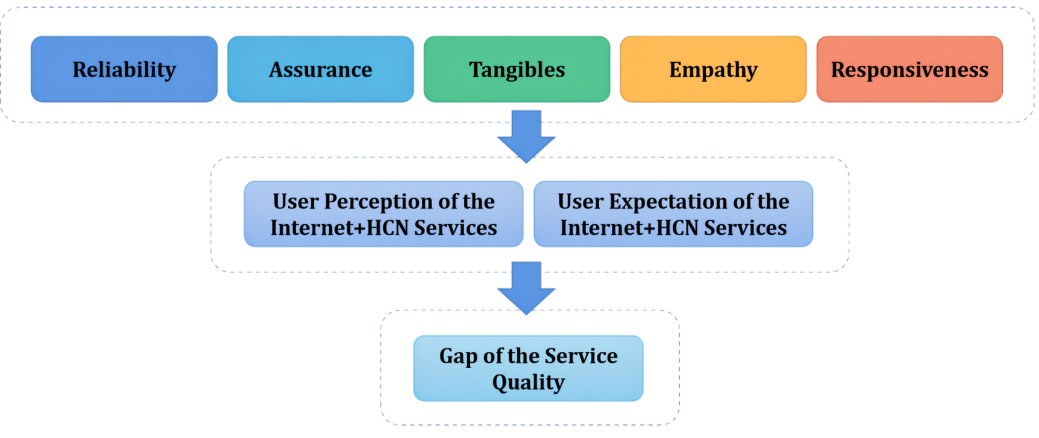

**Figure 1** Theoretic framework of the quality indicator system for evaluating the Internet+ HCN services.

survey. Efforts were made to recruit a panel that was both homogenous, when necessary for the study's objectives, and heterogeneous, to ensure a broad range of perspectives. The aim was to strike a balance between achieving a deep understanding of the subject matter and reaching a consensus that could be generalized to a broader context. By employing these recruitment strategies, the researchers aimed to assemble a panel of experts who could contribute to the Delphi study with authority, credibility, and a variety of viewpoints. Totally, the present study recruited 17 nationwide experts through purposive sampling from relevant fields, such as nursing education, clinical nursing, health management, and health informatics.

## Delphi procedure

Consultation questionnaires were distributed to experts using email surveys. They consisted of three parts: Part 1 was the cover letter, introducing the research background, objectives, and methods. Part 2 is a survey of the demographic information of the consulted experts, such as their age, working years, and education background. Furthermore, experts were requested to rate their acquaintance towards the research theme using a 5-level scale from very unfamiliar to very familiar. The scores of judgment basis of each expert were measured by a 3-level scale from significant, medium, to low impact, exploring the level of impact from theoretical analysis, experience, knowledge of literature, and instinct to the judgment. Part 3 is the consultation survey—the main body of the consultation questionnaire.

In the survey, experts were requested to rate the 20 potential quality indicators on a 5-point Likert scale, with higher marks indicating a higher level of importance. Besides, experts had the option to indicate their revision opinions after rating each indicator. This ensures that they can fully express their professional advice according to their own professionalism and experience.

Informed consent forms were completed by all experts and the confidentiality of the information they provided was strictly protected throughout the whole process. The two

rounds of Delphi consultation questionnaires were emailed in October and November 2022. After the Delphi procedure, analytic hierarchy process (AHP) questionnaires were also emailed to further determine the weights of each final indicator. All questionnaires were completed online. Reminder emails were sent to experts when necessary. All 17 experts completed the 2-round Delphi consultation and AHP procedure. The project has been approved by the Ethics Review Committee of Zhejiang Hospital with the approval number 2022LSD(121K).

### Delphi round 1

In the first round, experts were requested to rate the five primary indicators and 15 secondary indicators on a 5-point-Likert scale and to give comments on modifying or adding new indicators.

### Delphi round 2

Results in the first round were reviewed and the indicators were revised according to experts' comments. Round 2 Delphi consultation was performed to reach an agreement among the consulted experts.

### Indicator selection

In both rounds, the average value, standard deviation, and coefficient of variation (CV) of the importance rating of each indicator were calculated (*Nasa, Jain & Juneja, 2021*; *McPherson, Reese & Wendler, 2018*). Indicators were selected according to the following criteria: (1) the mean rating is no less than 4.0 and (2) the CV is less than 0.25. Thus, indicators with mean rating of importance less than 4.0 or a CV larger than 0.25 were deleted or modified according to experts' suggestions.

### AHP procedure

After the 2-round Delphi consultation, the quality indicators were finalized. All experts were invited to confirm the results and complete the AHP survey to form the comparison matrix of each indicator. AHP is a technique that assigns priorities on each alternative by identifying the importance of attributes hierarchically. Consequently, a comparatively priority weight was calculated for each indicator in the hierarchy structure.

The AHP procedure consists of two steps: Step 1 involves constructing the hierarchy structure, which is composed from the low to high levels based on secondary indicators, primary indicators, and the general goal. Here, the hierarchy refers to the quality indicator system for evaluating Internet+ HCN Services. Step 2 involves forming the comparison matrix. In this step, indicators at the same level are paired for comparison. For each pair of indicators, experts were requested to indicate the relative importance using the comparison scale, also known as the SAATY scale. In the SAATY scale, the comparisons between elements use a 1–9 scale and its reciprocals to assign values, representing the importance of one element over another in relation to the higher-level factor. In this study, attributing values that vary from 1 to 9, the scale determines the relative importance of an quality indicator when compared with another quality indicator. An odd number in the

scale was used to ensure a reasonable distinction among the measurement points, which means extreme importance (nine points), very strong importance (seven points), strong importance (five points), moderate importance (three points), and equal importance (one point) separately. An even number was adopted when consensus cannot be reached and the middle point served as a compromise. On this basis, a judgement matrix was built based on the comparison by the experts. The rating results were quantified and then normalized to develop a weight set (*Ing, 2022*).

## Data analysis

Data were entered and managed by Excel 2019. Statistical analysis was performed by SPSS 25.0. The general information and the degree of authority of experts were described by description analysis. The concentration of expert opinions was measured by the mean and standard deviation of the importance ratings for each indicator. The degree of consensus among experts was assessed using the coefficient of variation and Kendall's coefficient of concordance (Kendall's W), with a significance level of $\alpha = 0.05$. The higher values indicating better concordance. The degrees of authority (authoritative coefficients, Cr) of experts were determined by averaging the score of judgment basis (Ca) and their acquaintance with the research theme (Cs). YAAHP software was used to calculate the comparative weights in the AHP procedure. Statistical significance was set at $P < 0.05$.

# RESULTS

## General information and authority coefficients of experts

Altogether, 17 experts completed the two-round Delphi consultation and the AHP procedure. The response rates in both rounds of Delphi and the AHP were 100%. Table 1 shows the detailed general information of the experts.

The average expert authority coefficients were 0.912 and 0.925 respectively in Delphi rounds 1 and 2. Table 2 illustrates the comparison of degree of authority in two rounds.

The Kendall's W, indicating the coordination of experts' opinions, ranged from 0.262 to 0.265, which is displayed in Table 3. The value of variation coefficients is more acceptable in round 2.

## Indicator modifications

After each round of Delphi consultation, the research group held discussions to select and revise indicators according to the selection criteria and experts' suggestions.

In Delphi round 1, all indicators were eligible for the selection criteria (see Tables 4 and 5) and revisions were mainly made toward definitions of secondary indicators (degree of platform support, level of homogenization among providers, rational of the service plan, qualification of the agent, ability to provide personalized services).

In Delphi round 2, further revisions were made toward the ability to provide personalized services. Finally, five primary indicators, 15 secondary indicators, and their definitions were determined in the 2-round Delphi consultation (see Tables 6 and 7), followed by the AHP procedure.

**Table 1  General information of the experts.**

| Item | N | Proportion |
|---|---|---|
| Age (year) | | |
| 30–39 | 2 | 11.76% |
| >=40 | 15 | 88.24% |
| Working years | | |
| 11–20 | 2 | 11.76% |
| ≥20 | 15 | 88.24% |
| Education background | | |
| Bachelor | 9 | 52.94% |
| Master | 4 | 23.53% |
| Doctor | 4 | 23.53% |
| Professional titles | | |
| Medium-grade | 3 | 17.65% |
| Senior-grade | 14 | 82.35% |
| Specialized fields | | |
| Nursing education | 4 | 23.53% |
| Clinical nursing | 5 | 29.41% |
| Health management | 5 | 29.41% |
| Health informatics | 3 | 17.65% |

**Table 2  Degree of expert authority in two rounds.**

| Round | Judgement coefficient (Ca) | Familiarity coefficient (Cs) | Authority coefficient (Cr) |
|---|---|---|---|
| Delphi round 1 | 0.967 | 0.857 | 0.912 |
| Delphi round 2 | 0.925 | 0.925 | 0.925 |

**Table 3  Coordination degree of expert opinions.**

| Item | Delphi round 1 | | | Delphi round 2 | | |
|---|---|---|---|---|---|---|
| | Kendall's W | $X^2$ | P | Kendall's W | $X^2$ | P |
| Primary indicator | 0.337 | 22.906 | <0.001 | 0.232 | 14.839 | <0.001 |
| Secondary indicator | 0.256 | 60.873 | <0.001 | 0.280 | 62.815 | <0.001 |
| Total | 0.262 | 84.565 | <0.001 | 0.265 | 80.514 | <0.001 |

## Results of AHP

Based on the finalized quality indicator system, a judgement matrix was built according to comparisons of each pair of indicators by experts. Figure 2 shows the results, in which the rating values were quantified and normalized to develop a weight set. Each indicator received a calculated comparatively priority weight in the hierarchy structure.

**Table 4 Results of expert correspondence on primary indicators in Delphi rounds 1 and 2.**

| Primary indicator | Delphi round 1 | | Delphi round 2 | |
|---|---|---|---|---|
| | Importance (x ± s) | CV | Importance (x ± s) | CV |
| Reliability | 4.938 ± 0.25 | 0.051 | 4.941 ± 0.243 | 0.049 |
| Assurance | 4.813 ± 0.403 | 0.084 | 4.824 ± 0.393 | 0.081 |
| Tangibles | 4.5 ± 0.516 | 0.115 | 4.529 ± 0.514 | 0.114 |
| Empathy | 4.438 ± 0.512 | 0.115 | 4.471 ± 0.514 | 0.115 |
| Responsiveness | 4.75 ± 0.447 | 0.094 | 4.765 ± 0.437 | 0.092 |

**Table 5 Results of expert correspondence on secondary indicators in Delphi rounds 1 and 2.**

| Primary indicator | Secondary indicator | Delphi round 1 | | Delphi round 2 | |
|---|---|---|---|---|---|
| | | Importance (x ± s) | CV | Importance (x ± s) | CV |
| Reliability | Rational of the service plan | 4.813 ± 0.403 | 0.084 | 4.824 ± 0.393 | 0.081 |
| | Guaranteed service execution | 4.875 ± 0.342 | 0.070 | 4.882 ± 0.332 | 0.068 |
| | Proper presentation of the service records and feedback | 4.375 ± 0.5 | 0.114 | 4.412 ± 0.507 | 0.115 |
| Assurance | Qualification of the agent | 4.875 ± 0.342 | 0.070 | 4.882 ± 0.332 | 0.068 |
| | Knowledge and skill of the provider | 4.875 ± 0.342 | 0.070 | 4.882 ± 0.332 | 0.068 |
| | Attitude of the provider | 4.6 ± 0.507 | 0.110 | 4.625 ± 0.5 | 0.108 |
| | Ability to provide on-site support | 4.867 ± 0.352 | 0.072 | 4.875 ± 0.342 | 0.070 |
| Tangibles | Degree of platform support | 4.625 ± 0.5 | 0.108 | 4.647 ± 0.493 | 0.106 |
| | Intensity of platform attention | 4.375 ± 0.619 | 0.142 | 4.412 ± 0.618 | 0.140 |
| | Level of homogenization among providers | 4.313 ± 0.602 | 0.140 | 4.353 ± 0.606 | 0.139 |
| | Consistency of service and demand | 4.938 ± 0.25 | 0.051 | 4.941 ± 0.243 | 0.049 |
| Empathy | Ability to provide personalized services | 4.533 ± 0.516 | 0.114 | 4.563 ± 0.512 | 0.112 |
| | Give priority to the interests of the clients | 4.467 ± 0.743 | 0.166 | 4.375 ± 0.806 | 0.184 |
| Responsiveness | Responsiveness of routine service requirements | 4.688 ± 0.479 | 0.102 | 4.706 ± 0.47 | 0.100 |
| | Active response to temporary service requirements | 4.188 ± 0.75 | 0.179 | 4.875 ± 0.342 | 0.174 |

**Table 6 Definition of primary indicators.**

| Primary indicator | Definition |
|---|---|
| Reliability | The ability of the Internet+ HCN service to provide reliable services, which accurately meet user's service demand, and the ability to guarantee service execution. |
| Assurance | The relevant ability of the platform, agent, and provider to gain trust from the service user; for example, provider's knowledge, skill, and attitude. |
| Tangibles | Service-related facilities and circumstances are directly perceived by service users; for example, the information platform, service provider's personal image, and performance. |
| Empathy | The ability of the platform, agent, and provider to attract the users. |
| Responsiveness | The ability of responding to users' service demands. |

**Table 7 Definition of secondary indicators.**

| Primary indicator | Secondary indicator | Definition |
|---|---|---|
| Reliability | Rational of the service plan | Once ordered, a correct service plan is activated. The assigned HCN service provider is qualified with a proper professional certification that is in line with the active service plan. |
| | Guaranteed service execution | The implementation of the HCN service is consistent with its description. In case of emergency, the assigned HCN service provider could offer emergency advice, and perform necessary and prompt rescue or transfer measures. |
| | Proper presentation of the service records and feedback | Once complete, the service records and feedback are properly presented. |
| Assurance | Qualification of the agent and the service provider | Qualification of the agent and staffs are indicated in the platform. |
| | Knowledge and skill of the service provider | The assigned HCN service provider is proficient in the knowledge and skill required by the service task. |
| | Attitude of the service provider | The assigned HCN service provider performs professional communication. |
| | Ability to provide on-site support | In case of emergency, the assigned HCN service provider could seek help through the platform. |
| Tangibles | Degree of platform support | The degree of platform support in view of the user. |
| | Intensity of platform attention | The intensity of platform attention in view of the user. |
| | Level of homogenization among providers | The level of homogenization among providers in view of the user, such as wearing a unified professional uniform and following the same operating procedures. |
| | Consistency of service and demand | The consistency of service and demand in view of the user. |
| Empathy | Ability to provide personalized services | The ability to provide personalized services in view of the user. |
| | Give priority to the interests of the clients | The ability to give priority to the interests of the clients in view of the user. |
| Responsiveness | Responsiveness of regular service requirements | The ability to respond to regular and routine service requirements. |
| | Active response to temporary service requirements | The ability to actively respond to temporary service requirements. |

With regard to primary indicators, the weights of tangibility, reliability, responsiveness, assurance, and empathy were 0.192, 0.240, 0.190, 0.245, and 0.132, respectively. Among them, Reliability (0.240) and Assurance (0.245) dominated the proposed quality indicator system.

For the secondary indicators, weights ranged from 0.041 to 0.107. The top three most important indicators were Responsiveness of routine service requirements (0.107), Guarantee of the service execution (0.094), and Active response to temporary service requirements (0.083).

## DISCUSSION

### Composition and weight of the proposed quality indicator system

Internet+ HCN services are composed of the execution of the service provider and the experience of the user. The quality of Internet+ HCN services is affected by the service provider's skills and working performance, as well as the user's perception and differential demand. Consequently, the quality evaluation of Internet+ HCN services should take into account the objective situation of the service and the subjective feelings of the users.

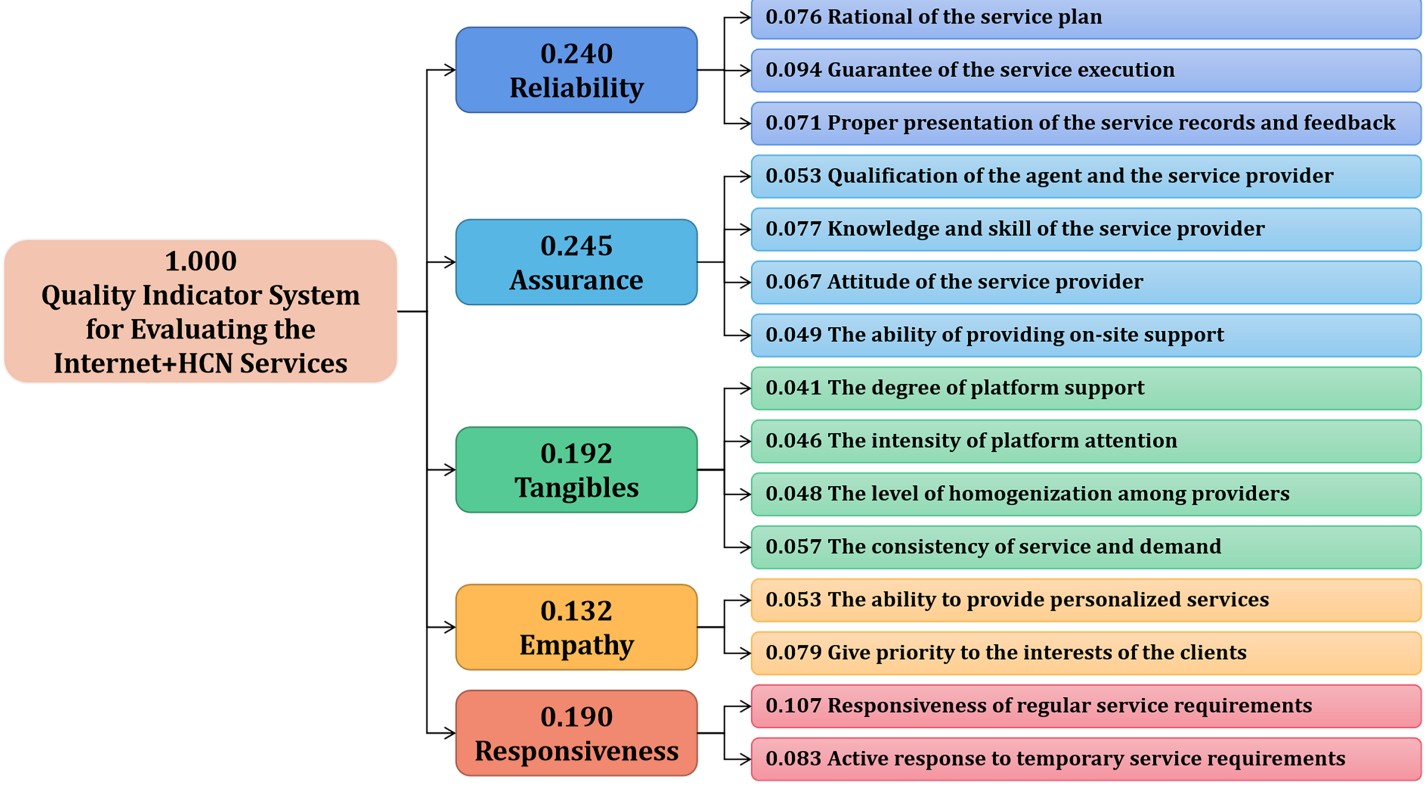

**Figure 2 Weight set of the quality indicator system for evaluating Internet+ HCN services from AHP.**

This study is innovative to some extent as it has enriched the humanistic nature of service quality evaluation and has profoundly mined users' services demands.

Experts reached a consensus on the primary and secondary indicators in Delphi round 1, and the vast majority of their suggestions revolved around the definition of secondary indicators.

For the primary indicators, all five dimensions of the RATER model, a smaller version of the SERVQUAL model, were utilized: reliability, assurance, tangibles, empathy, and responsiveness. Each primary indicator corresponds to two to four secondary indicators. In total, 15 indicators including platform agent, implementation guarantee, meet demand, regular demand, and interest protection, were determined. The five primary indicators are as follows from highest to lowest weight: assurance (0.245), reliability (0.240), tangibles (0.192), responsiveness (0.190), and empathy (0.132). The findings help determine the priority considerations in service quality and provided scientific guidance for the high-quality development Internet+ HCN services.

## Strengths of the proposed quality indicator system

Previous researches have provided empirical evidence of the effectiveness of internet-based intervention. And the current study offers complementary perspectives on the quality evaluation of internet-based interventions in healthcare, particularly in the home care

nursing domain. It delves into the development of a quality evaluation framework tailored to the unique characteristics of Internet+ HCN services.

As Internet+ HCN services are still in the early stages of development worldwide, there is no universally accepted indicator system in its quality evaluation (*Courtney et al., 2005*; *Wagner et al., 2020a*, *2020b*). Some developed countries such as America and some European countries have been launching home care nursing services for a relatively long time. Most of these services are supported by health insurance. In those countries, institution-led case management in home care nursing services are predominant (*Sinn et al., 2022*; *Kajander-Unkuri et al., 2022*). Consequently, comprehensive home care quality indicators are usually adopted in assessing the service quality of in-home care nursing (*Zanaboni et al., 2018*; *Busnel, Vallet & Ludwig, 2021*), which cover home care-related clinical, function, social, and service indicators as a whole (*Portea, 2021*).

The methods for evaluating "Internet+" and home care nursing services have been preliminarily explored in previous studies. However, existing studies on internet plus services quality evaluation have limitations in both width and depth, as they do not fully consider the characteristics of internet plus and lack evaluative dimensions from user perspectives (*Ma et al., 2022*). In China, some studies have proposed several quality evaluating systems for Internet+ nursing, but they do not specifically focus on home care nursing. Most of these studies have developed a system based mainly on the Donabedian model (*Wang et al., 2021a*; *Ren et al., 2022*; *Tan et al., 2022*; *Zhuang et al., 2021*; *Zhan et al., 2022*; *Donabedian, 2005*). Most of these studies have developed systems of indicators according to Donabedian's Structure-Process-Outcome model, which does not focus on users' experience.

This study considers the priority objective of service, which is to meet the demand of the user. Compared with a structure-process-outcome framework based quality evaluation system (*Tan et al., 2022*; *Zhan et al., 2022*), the proposed system facilitates an effective evaluation with real-time feedback based on user experience.

This study offers a dynamic evaluation within a time span compared with conventional evaluation models characterized by static evaluation, and facilitates the continuous quality improvement by user-perspective evaluation. In traditional home care nursing services, comprehensive home care quality indicators are adopted in assessing service quality (*Zanaboni et al., 2018*; *Busnel, Vallet & Ludwig, 2021*), such as the RAI-HC (*Portea, 2021*). Such indicators focus more on service related outcomes, such as activities of daily living, incontinence, fall, and pain in measuring service quality level (*Courtney et al., 2005*; *Bowles et al., 2021*). In this study, an in-depth exploration of service demand was conducted from the user perspective, avoiding the risk of neglecting user demand when focusing more on outcome related indicators (*Zhuang et al., 2021*). The proposed Internet+ HCN quality evaluation system incorporates service-centered evaluation indicators, which could strengthen the quality monitoring of service structure and process from the user perspective.

Besides, the proposed quality evaluation system fits perfectly with the three phases of the Internet service transaction. In the pre-implementation phase, fundamental hardware

and software guarantees, including services platforms, services staff qualifications, and technical support, constitute the basis condition for achieving high-quality Internet+ HCN services (*Wiener, Segelman & White, 2020*). In the implementation phase, factors related to service reliability, such as rational services schedule, services execution guarantees, services records, and acceptable feedback, play key roles in optimizing a user's service experience. In the last phase of post-implementation, factors related to service tangibility and responsiveness are vital (*Wang et al., 2021a*).

## LIMITATIONS

While Internet+ HCN services are still being piloted in China and the market is yet to mature, they are already expanding rapidly. Rooted in the perspective of user experience, the present study developed a comprehensive, scientific, and practical evaluation system for assessing the quality of Internet+ HCN services. The proposed system reflects the overall quality of the service in a systematic way, promoting continuous improvement of the service induced by the user side.

Due to exposure restrictions, the study only invited experts from mainland China who majored in related fields. Further research is thus needed before introducing the evaluation system to real care. As a next step, the constructed quality indicator system for evaluating Internet+ HCN services can be used for patients with validation and effect evaluation.

## CONCLUSION

The development and implementation of the proposed Internet+ Home Care Nursing (HCN) quality indicator system represents a significant advancement in the evaluation of this emerging service model. By incorporating both objective service-related indicators and user-centric subjective experiences, the system offers a comprehensive and dynamic framework for assessing and improving the quality of Internet+ HCN services. The system prioritizes user needs and experiences, ensuring that the focus remains on the primary goal of meeting patient demands and improving health outcomes. It balances the evaluation of service structure and process with user experience, providing a more holistic understanding of service quality. In the future, the system should be further validated and tested in diverse real clinical settings, and should be explored in ways to integrate the quality indicators with existing clinical data systems for more efficient management. By addressing these areas for further development, the proposed quality indicator system can serve as a valuable tool for healthcare providers, policymakers, and researchers to ensure the quality and effectiveness of Internet+ HCN services and ultimately improve patient care.

## ACKNOWLEDGEMENTS

The authors are grateful to undergraduate student, Xiaoyi Jiao for her suggestions on the revision of the article.

### Funding

This work was supported by the National Key R&D Program of China (2019YFE0113100). The funders had no role in study design, data collection and analysis, decision to publish, or preparation of the manuscript.

### Grant Disclosures

The following grant information was disclosed by the authors:
National Key R&D Program of China: 2019YFE0113100.

### Competing Interests

The authors declare that they have no competing interests.

### Author Contributions

- Lei Ye performed the experiments, analyzed the data, prepared figures and/or tables, authored or reviewed drafts of the article, and approved the final draft.
- Shulan Yang conceived and designed the experiments, analyzed the data, prepared figures and/or tables, authored or reviewed drafts of the article, and approved the final draft.
- Biyan Jiang performed the experiments, authored or reviewed drafts of the article, and approved the final draft.
- Caixia Liu performed the experiments, analyzed the data, authored or reviewed drafts of the article, and approved the final draft.
- Xiaoqing Jin analyzed the data, authored or reviewed drafts of the article, and approved the final draft.
- Polun Chang conceived and designed the experiments, authored or reviewed drafts of the article, and approved the final draft.

### Human Ethics

The following information was supplied relating to ethical approvals (*i.e.*, approving body and any reference numbers):

The project has been approved by the Ethics Review Committee of Zhejiang Hospital with the approval number 2022LSD(121K).

### Data Availability

The raw measurements are available in the Supplemental File.

### Supplemental Information

Supplemental information for this article can be found online at http://dx.doi.org/10.7717/peerj.18281#supplemental-information.

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
