# Peer review of "Developing a quality indicator system for evaluating internet plus home care nursing services based on the SERVQUAL model: a Delphi-analytic hierarchy process study"

_PeerJ, doi:10.7717/peerj.18281_

## Round 0.1 · original submission · Major Revisions

This is an interesting and relevant topic. There are, however, a number of revisions required to improve this manuscript. The reviewers have highlighted a number of key areas that require revision. In particular, the background and discussion sections require more detail and expansion. For your background, please provide in greater detail:
- how your sample of experts was recruited (targeted, advertising etc)
- how the evaluated quality indicators were developed
- how the literature search described at line 125 was undertaken and what the results were
- background to the SERVQUAL, RATER and SAATY scale

In your data analysis, please detail what is meant by the 'concentration' analysis.

The discussion section requires more context of your results in the setting of current literature.

The limitations section should be separate to your conclusions.

Reviewer 1 ·

Basic reporting

Abstract/Introduction/Background
The problem of practice is not clearly presented and its importance to practice is not identified. There are very limited attempts to describe the relationships between current literature, policy, and practice. The background is very limited.
The aims and objectives of the review are not clearly identified.
A very weak argument that the review contributes to knowledge in the subject is presented.

Literature review/Background (before your methodology)
Not existent.
Does not discuss status or gaps in current research literature. The literature review is entirely or virtually descriptive, showing little or no evidence of appropriate analysis of the significance of the relationship to the study Most important/current literature has not been identified/explored. No evidence of wide reading Shows an insufficient level of factual and conceptual understanding of the subject appropriate to their research question. Has accepted information uncritically. Unsubstantiated opinions are common. Literature review does not establish a theoretical framework.

Methodology:
Presents a methodology that has flaws in its suitability to the proposed research and/or its design is nonsystematic/ not fully considered. Some areas of the proposed study are discussed (e.g. sampling strategy, tool selection/ development, recruitment, data collection and analysis, ethical issues, reliability/ validity but for the most part uncritically and with significant omissions.

Findings:
The findings of the research are presented in a manner which demonstrates very limited understanding of either the process of the methodology used and the implications of the findings for nursing practice. The method of analysis is suitable but the findings briefly presented. Inadequate knowledge and understanding is identified from the study, the contribution of this to enhancing nursing practice is not clearly articulated.

Discussion:
Not presented

Presentation:
Old references: Parasuraman, Zeithaml & Berry, 1985; for example. Make sure to use references no older than 10 years.
The order in there references seems random. For example: Marc et al., 2019; Abdi et al., 2019; Keevil & Romero-Ortuno, 2015; Aaltonen & van Aerschot, 2021. It would make more sense use the references in a specific order.
Good use of pictures and tables

Experimental design

no comment

Validity of the findings

no comment

Reviewer 2 ·

Basic reporting

Dear Colleague,
I am honored that you chose me to review the article. I hope you will be satisfied with my review of the article and the answer.
I am sending my opinion about the article below.

General comments
The title of the article "Developing a quality indicator system for evaluating internet plus home care nursing services based on the SERVQUAL model: a Delphi-analytic hierarchy process study" is very interesting and points to an interesting topic. The title properly explains the purpose and objective of the article.
The abstract of the article is structured and contains the necessary parts: background, objective, methods, results, conclusions and keywords. The abstract contains an appropriate summary of the article, the language used in the abstract is easy to read. Authors do provide adequate background on the topic and reason for this article and describe what the authors hoped to achieve.
In the introduction, the authors clearly and succinctly describe the problems of demographic change in China. More specifically, the authors describe the increased proportion of elderly people in China and the need for new approaches in elderly care.
The research design is described in detail.
The research design is appropriate and does not contain particular weaknesses.
The measurement instrument is clearly described.
The population of interest and sampling procedure are clearly defined.
The data collection procedure is clearly described.
The data analysis is stated in precise terms (Delphi procedure 1 and 2)
Results: the results are presented clearly, the authors provide accurate research results, and their is sufficient evidence for each result.
Tables and figures are used efficiently.
Conclusion: I suggest that the authors expand the conclusion accordingly to the results. In the conclusion, the authors mostly state the limitations of the study.
Finally, this was an interesting article information to the filed. In its current state, it adds much new insight.


Basic reporting
The article is written in professional English. ). I suggest that the authors correct the text n 300-301: ).-Most of these studies have developed a system of indicators according to Donabedian's structure-process-outcome quality of care model, which does not focus on users' experience. (Donabedian, 2005; Ma et al., 2022), which does not focus on users' experience.

Literature references, sufficient field background/context.
The article contains enough introduction and background to show how the paper fits into the wider body of knowledge. Relevant prior literature is cited appropriately.

Expert structure of the article, figures ( 2 ), tables (8).

The structure of the article is in accordance with the acceptable format of 'standard sections'. '

The figures are relevant to the content of the article, have sufficient resolution and are adequately described and labeled.

All relevant raw data are made available.

Experimental design

Original primary research within the goals and scope of the journal.
Research question well defined, relevant and meaningful. It is stated that the research fills the identified knowledge gap in the care of the elderly.
The research question is a clear question. The authors have identified a knowledge gap under investigation and should make statements about how the study contributes to filling that gap. The authors also stated the limitations of the study.

Rigorous investigation conducted according to high technical and ethical standards.
The research was conducted in accordance with the applicable ethical standards in the field.

Methods described with sufficient detail and information for replication.
The methods are described with sufficient information so that another examiner can repeat them.

Validity of the findings

All basic data are listed; they are robust, statistically reliable.
The data on which the conclusions are based are available in an acceptable repository.

Conclusion: I suggest that the authors expand the conclusion accordingly to the results. In the conclusion, the authors mostly state the limitations of the study.

---

## Round 0.2 · Minor Revisions

Thank you for submitting a revised version of your manuscript. There are a few outstanding minor issues that require addressing:

- please provide an appropriate academic reference for the RATER scale and a rationale for why this was chosen.
- please remove the term 'systematic' from your literature search, as you have not undertaken a systematic literature review.
- please provide further information about how many manuscripts were reviewed as part of your literature review and what potential indicators emerging from these results were. It would be helpful to discuss how these findings were then integrated into your interviews with the expert panel.
- Are you able to provide the interview guide for your semi structured interviews as a supplemental file? It is stated that this was with service providers and users - more information about this is required e.g. how many were interviewed, how were results analysed etc.
- There are grammatical and typographical errors throughout the manuscript that require correcting.

Reviewer 2 ·

Basic reporting

The authors corrected the text and corrected errors according to the suggestions of reviewers and editors.

Experimental design

The authors corrected the text and corrected errors according to the suggestions of reviewers and editors.

Validity of the findings

The authors corrected the text and corrected errors according to the suggestions of reviewers and editors.

---

## Round 0.3 · accepted · Accept

Thank you for addressing the remaining minor comments. There remain some grammatical issues, which can be corrected prior to publication. I believe the interview schedule you have presented should be included as a supplemental file.